# Growth and Biochemical Composition of Microgreens Grown in Different Formulated Soilless Media

**DOI:** 10.3390/plants11243546

**Published:** 2022-12-15

**Authors:** Roksana Saleh, Lokanadha R. Gunupuru, Rajasekaran Lada, Vilis Nams, Raymond H. Thomas, Lord Abbey

**Affiliations:** 1Department of Plant, Food, and Environmental Sciences, Faculty of Agriculture, Dalhousie University, 50 Pictou Road, Bible Hill, NS B2N 5E3, Canada; 2Biotron Experimental Climate Change Research Centre, Department of Biology, University of Western Ontario, London, ON N6A 5B7, Canada

**Keywords:** microgreens, natural amendment, soil health, phytochemicals, healthy food

## Abstract

Microgreens are immature young plants grown for their health benefits. A study was performed to evaluate the different mixed growing media on growth, chemical composition, and antioxidant activities of four microgreen species: namely, kale (*Brassica oleracea* L. var. *acephala*), Swiss chard (*Beta vulgaris* var. *cicla*), arugula (*Eruca vesicaria* ssp. *sativa*), and pak choi (*Brassica rapa* var. *chinensis*). The growing media were T1.1 (30% vermicast + 30% sawdust + 10% perlite + 30% PittMoss (PM)); T2.1 (30% vermicast + 20% sawdust + 20% perlite + 30% PM); PM was replaced with mushroom compost in the respective media to form T1.2 and T2.2. Positive control (PC) was Pro-mix BX™ potting medium alone. Root length was the highest in T1.1 while the shoot length, root volume, and yield were highest in T2.2. Chlorophyll and carotenoid contents of Swiss chard grown in T1.1 was the highest, followed by T2.2 and T1.1. Pak choi and kale had the highest sugar and protein contents in T2.2, respectively. Consistently, total phenolics and flavonoids of the microgreens were increased by 1.5-fold in T1.1 and T2.2 compared to PC. Antioxidant enzyme activities were increased in all the four microgreens grown in T1.1 and T2.2. Overall, T2.2 was the most effective growing media to increase microgreens plant growth, yield, and biochemical composition.

## 1. Introduction

Microgreens are immature greens harvested from tender young plants that are grown for their high health-promoting compounds and biological properties [1,2]. Previous researchers reported high amounts of phytochemicals such as ascorbic acid, α-tocopherol, β-carotene, phylloquinone, vitamins, and minerals in different species of microgreens [3,4,5]. Kale (*Brassica oleracea* L. var. *acephala*), Swiss chard (*Beta vulgaris* var. *cicla*), and arugula (*Eruca vesicaria* ssp. *sativa*), as microgreens, possess high levels of vitamins A, C, and K, essential lipids, carotenoids, and mineral nutrients [5,6]. Microgreens are delicate and are prone to various stress factors that can adversely affect the edible quality and bio-functional properties. Like all plants, the key preharvest factors that can affect a microgreen’s edible quality are genotypic characteristics, growing media, climate, and management practices [7,8,9,10]. Hence, the presented study focuses on the impact of various growing media amendments on the quality of different microgreens. Natural amendments are organic substrates added to a growing medium to improve plant productivity and harvest quality, through enhancement of the physiochemical properties and functional activities of the media [11,12,13]. These amendments include compost, vermicast, humates, manures, and sawdust. They supply macro- and micro-nutrients, support beneficial microbes, improve water-holding capacity and gas exchange, and promote nutrient availability required for plant growth and development [13,14,15].

Vermicast (earthworm excreta or castings) is a humus-like material rich in beneficial microbiome and humic and non-humic substances such as mineral elements, amino acids, plant hormones, and other macromolecules that promote to plant growth and development [16,17]. According to Karthikeyan et al. [18], vermicast enhanced the seed germination rate and plant growth parameters, leaf pigmentation, root nodulation, and the yield of Lantana (*Lantana camara*) and cluster bean (*Cyamopsis tetragonoloba*), compared to inorganic fertilizer. In addition, adding vermicast to a growing media ameliorated soil physiochemical properties, leading to improved aeration, media porosity, field capacity, and microbial activity [19,20]. Similarly, Abbey et al. [14] showed that morphological indices of kale and postharvest essential fatty acids, mineral nutrients, phenolic compounds, and antioxidant capacity were increased by the application of dry vermicast, potassium humate, and volcanic minerals.

Sawdust is another potential growing medium substrate that is a waste from the forestry and wood industries. Currently, sawdust is burned or taken to landfills. There is a growing concern over the mining and use of Sphagnum peat moss. Therefore, sawdust can be used as an environmentally friendlier alternative or supplement to traditional substrates such as peat moss or can be used in combination with other substrates. Maharani et al. [21] showed that sawdust can improve the porosity and drainage of a growing medium. A study showed that sawdust delayed the initial growth of tomato seedlings (*Solanum lycopersicum*), but that the plant growth soared seven weeks after planting when the seedlings were established, and that the yield was higher than that of the control [22]. This delay can be attributed to toxic compounds from the wood such as lignin, cellulose, hemicellulose, and terpenes, which probably leached out, decomposed, or diluted by reaction with other amendments after seven weeks of planting [22,23,24].

Chang [25] showed that the combination of sawdust with 30% soil, plus nitrogen (N), phosphorus (P), and potassium (K) compound fertilizers gave rise to a higher productivity of the tomato plant compared to sawdust alone. Plant growth components, yield index, and nutritional values of *Syngonium podophyllum* were drastically increased following the application of vermicompost-sawdust extract [26]. A recent study showed that the combination of different proportions of vermicast and sawdust improved plant growth and biochemical compounds in Swiss chard, pak choi, and kale microgreens [23]. The authors found that 40% vermicast + 60% sawdust, or 60% vermicast + 40% sawdust improved the physiochemical properties of the growing media and enhanced the active microbial activity and nutrient mineralization necessary to meet potential plant growth requirements.

Therefore, amendments such as vermicast and compost can be added to sawdust to improve both the nutrient status and functionality of the growing medium. A study by Hernánde et al. [27] showed that the application of spent mushroom compost increased the seed germination percentage, fresh shoot weight, and yield of red baby leaf lettuce (*Lactuca sativa* L.) by up to 7-fold, compared to peat alone. Few studies on the effects of individual amendments on plants have been reported, but not on their combining effect on microgreen plant growth and chemical composition. Therefore, the objective of the present study was to evaluate the physiochemical properties of different proportions of mixed media and their effects on the growth and biochemical composition of four different plant species (kale, Swiss chard, arugula, and pak choi) that can be grown and harvested as microgreens.

## 2. Results

### 2.1. Growing Media Properties

The different additives in the growing media significantly affected the physiochemical properties (Table 1). It was found that T1.1 and T2.1 had a significantly (*p* < 0.05) low bulk density of an average of 0.07 g/cm^3^ compared to an average of 0.10 g/cm^3^ for T1.2, T2.2, PC, and NC. The highest porosity was observed in PC, followed by T1.2, and T2.2 compared to the other treatments. Porosity and field capacity of media T1.1, T2.1, and NC were significantly (*p* < 0.05) lower than the other media.

The different growing media had pH values ranging from 5.7 to 6.4. The pH for T1.1 was significantly (*p* < 0.05) lower than that of T1.2. The overall trend for salinity, electrical conductivity, and total dissolve solids of the growing media was similar among the treatments (Table 1). NC had the highest salinity, electrical conductivity, and total dissolved solids followed by T1.2, then T1.1, and T2.1, and the least by NC.

### 2.2. Plant Growth and Yield

The growing media, plant species, and the interaction of growing media × plant species influenced plant growth components significantly (*p* < 0.01).

Total root lengths of arugula, pak choi, kale, and Swiss chard were increased by ca.79%, 83%, 61%, and 62% in T1.1, respectively, compared to the average for their counterparts grown in the PC and NC (Figure 1A). T1.1, T2.1, and T2.2 similarly had the highest effect on total root length compared to the others. Total shoot length of arugula, pak choi, kale, and Swiss chard were increased by ca. 99%, 105%, 62%, and 115%, respectively, in T2.2, compared to their counterparts in the PC (Figure 1B).

Consistently, the PC and the NC significantly (*p* < 0.01) reduced the total length of the roots and shoots of all the microgreen plants. Furthermore, the root volume was increased by ca. 67% to 143% in plants grown in T2.2, compared to those grown in the PC (Figure 1C). Consistently, the root volume of each plant was significantly (*p* < 0.01) reduced in T1.1, followed by NC and then PC (Figure 1C). The plant yield of the microgreens was significantly (*p* < 0.01) increased by ca. 230% in T2.2 and 160% in T1.2, respectively, compared to their PC counterparts (Figure 1D). Consistently, PC and T1.1 significantly (*p* < 0.01) reduced the yield of all the microgreens.

### 2.3. Microgreens Biochemical Composition

The ANOVA demonstrated that variations in the mixed media, plant species, and their interaction, significantly (*p* < 0.01) affected the biochemical compositions of the microgreens (Figure 2A–D). Total carotenoids, Chl a, Chl b, and Chl t of all the microgreens were increased significantly (*p* < 0.05) by T1.1 and T2.2, except Chl b in the pak choi, which was increased by T2.2 (Figure 2B). T1.2 had a similar effect to T1.1 and T2.2 in increasing Chl a, Chl b, Chl t and the total carotenoids in arugula and kale microgreens, but the effect varied for pak choi and Swiss chard (Figure 2A–D). Total chlorophyll and carotenoids were approximately 1.5-fold higher in T1.1 and T2.2 compared to their PC counterparts. Moreover, among the different plant species, kale and Swiss chard exhibited the highest Chl t by 67% in T1.2 and by 116% in T1.1 compared to PC (Figure 2C).

Likewise, the highest total carotenoid content was about 72% higher for both kale and Swiss chard in T2.2 and T1.1, respectively, compared to their PC counterpart (Figure 2D). The total carotenoid content of arugula and pak choi was increased by ca. 15% and 24% in T2.2, respectively, compared to plants grown in the PC. Consistently, the lowest total carotenoid content was observed in all the microgreens grown in the T2.1, except for kale, which was lowest in the PC (Figure 2D). The overall trend for total carotenoid was arugula (562.35 µg/g FW) > Swiss chard (518.02 µg/g FW) > kale (472.69 µg/g FW) > pak choi (391.68 µg/g FW) (Figure 3D).

The highest sugar content was recorded by arugula microgreens grown in the PC, followed by T2.2 compared to other treatments (Figure 3A). On the contrary, the sugar content of pak choi was increased by 73% in T2.2 while T1.1 increased the sugar content of kale and Swiss chard by ca. 23% and 65%, respectively, compared to the PC (Figure 3A). Consistently, T2.1 significantly (*p* < 0.01) reduced the sugar content of all the four different microgreens. Among the microgreen plant species, the overall trend for the sugar content was arugula (3624.40 μg glucose/g) > kale (3204.99 μg glucose/g) > pak choi (3118.44 μg glucose/g) > Swiss chard (1944.46 μg glucose/g) (Figure 3A). As shown in Figure 3B, T1.1 significantly (*p* < 0.01) increased the protein content in arugula and Swiss chard by ca. 37% and 55%, respectively; while T2.2 significantly (*p* < 0.01) increased the protein content in pak choi and kale by ca. 23% and 105%, respectively, compared to their counterparts grown in the PC. The other media had similar effects on the total protein content of the microgreens. Overall, the trend for the protein content was Swiss chard (6372.85 µg Bovine/g) > kale (4941.84 µg Bovine/g) > arugula (4782.70 µg Bovine/g) > pak choi (3901.83 µg Bovine/g) (Figure 3B).

Total phenolics were significantly (*p* < 0.01) increased in all the plants grown in T1.1, followed closely by T2.1, which were not significantly (*p* > 0.05) different for Swiss chard (Figure 3C). The increase in total phenolics in arugula, pak choi, and kale by T1.1 and T2.1 were on the average, 1.5- and 1.2-fold higher than their counterparts that were grown in the PC. Interestingly, Swiss chard, followed by pak choi, and then arugula and kale had phenolics contents of ca. 144%, 63%, 50%, and 29% in T1.1, respectively, compared to their counterparts that were grown in the PC. Comparatively, the trend for the phenolics content in the microgreens was arugula (241.76 mg GAE/g) > kale (180.08 mg GAE/g) > pak choi (169.18 mg GAE/g) > Swiss chard (151.44 mg GAE/g) (Figure 3C). Total flavonoids in all the microgreens grown in T2.2, except for Swiss chard, increased by 1.5-fold compared to the microgreens grown in PC (Figure 3D). Total flavonoids in Swiss chard increased by 51% in T1.1 compared to PC. Total flavonoids in arugula, kale, and pak choi increased by 65%, 56%, and 31%, respectively, in T2.2 compared to PC. Among the microgreen plant species, the overall trend for the flavonoid was Swiss chard (638.34 µg quercetin/g) > arugula (553.84 µg quercetin/g) > kale (362.50 µg quercetin/g) > pak choi (360.96 µg quercetin/g) (Figure 3D).

The total ascorbate was increased by 57%, 64%, and 51% in arugula, pak choi, and kale grown in T1.2, respectively, compared to PC (Table 2). Furthermore, Swiss chard ascorbate content was significantly (*p* < 0.01) increased by 83% and 73% in T2.2 and T1.2, respectively, compared to PC. On the contrary, ascorbate was significantly (*p* < 0.01) reduced in microgreens grown in the T2.1 (Table 2). The overall trend for the microgreens’ ascorbate content was kale (25.90 μmol/g FW) > Swiss chard (24.22 μmol/g FW) > arugula (23.41 μmol/g FW) > pak choi (22.40 μmol/g FW) (Table 2). Peroxidase was significantly (*p* < 0.01) increased in arugula and Swiss chard by T1.1 and T2.2 while T1.2 significantly (*p* < 0.01) increased POD in pak choi and kale.

Comparatively, Swiss chard followed by arugula had the highest POD activity and pak choi followed by kale had the lowest. The overall trend for the microgreens POD activity was pak choi (0.88 Unit/mg FW) > Swiss chard (0.66 Unit/mg FW) > arugula (0.58 Unit/mg FW) > kale (0.48 Unit/mg FW). Furthermore, APEX activity increased by 77% in Swiss chard and by 68% in kale when grown in T2.2, compared to PC. APEX activity of arugula and pak choi were increased by 55% and 54% in T1.1 and T1.2, respectively, compared to those grown PC (Table 2). Among the microgreen plant species, the overall trend for the microgreens APEX activity was arugula (0.146 Unit/mg FW) > Swiss chard (0.103 Unit/mg FW) = kale (0.103 Unit/mg FW) > pak choi (0.066 Unit/mg FW) (Table 2). Consistently, PC and NC significantly (*p* < 0.01) reduced the biochemical composition of the different microgreens (Table 2).

### 2.4. Association among Media, Plants, and Biochemical Composition

A multivariate two-dimensional PCA biplot was used to assess the association between the microgreens plant yield and biochemical parameters, as influenced by variations in growing media formulations (Figure 4). The PCA explained 80% of the total variations in the dataset. Treatments that are close to the origin of the PCA axes show a high association and stability than those on the periphery. The PCA demonstrated that treatment T2.2 can be associated with an improved plant yield and biochemical composition of the microgreens. The interaction of the growing media and plant species can be closely associated with the microgreens’ yield and total ascorbates. Furthermore, kale carotenoid content was strongly influenced by the interaction between the growing media × plant species compared to the other plant species. APEX activities were associated with the interaction between the growing media and plant species in all the microgreens except arugula (Figure 4). Overall, the interaction between the growing media and plant species can be associated with the kale yield and its biochemical parameters compared to the other plant species.

## 3. Discussion

The effects of different substrates on the physiochemical characteristics of the formulated growing media and the differential response of the four different microgreens plant species were investigated under greenhouse conditions. The growing medium T1.2, followed by T2.2, had the highest effect on most of the plant growth components, except for root length. The growing media T1.1 and T2.1 contained PittMoss, which was made from mainly shredded cardboard, and T1.2 and T2.2 contained mushroom compost. The results show that mushroom compost was more beneficial than PittMoss. Similarly, Renaldo et al. [28] reported that mushroom compost increased the shoot and root dry mass in cucumber (*Cucumis sativus*) compared to biochar and corn stalks but had no effect on lettuce (*Lactuca sativa*), probably due to lettuce intolerance of the high salt content in the mushroom compost. Furthermore, Vahid Afagh et al. [15] reported that a 15% mushroom compost mixed in sandy loam soil increased both the plant growth and yield of German chamomile (*Matricaria recutita* L.) due to the improved medium structure, increased nutrient availability, and beneficial microbial activity [15,29]. Furthermore, the results also suggested that the variations in response of the microgreen plants to the different media were dependent on genotypic differences.

It was obvious that the improved structure and functionality of growing media T1.2 and T2.2 improved plant growth in all the plant species except for pak choi, as previously explained by Emami and Astaraei [30] and Vahid Afagh et al. [15]. The root lengths of all the microgreens were significantly increased in T1.1 and T2.1 compared to the other media. According to Vahid Afagh et al. [15], an addition of 15% mushroom compost to a medium increased aeration and water-holding capacity, leading to an improved crop productivity. The addition of PittMoss in T1.1 and T2.1 reduced the growing media bulk density, which in turn promoted root growth compared to the mushroom compost. A previous study using a high bulk density of (i.e., 1.35 g/cm^3^) growing medium led to a reduction in lettuce root growth and yield [31]. In the present study, the bulk density ranged between 0.07 and 0.12 g/cm^3^, which was below the root-restriction threshold bulk density of 1.6 g/cm^3^, especially in T1.2 and T2.2. This may be the reason for the enhanced plant growth and yield of microgreens grown in T1.2 and T2.2. Moreover, Gillespie et al. [32] stated that the optimum range of pH for leafy greens growth is a 5.5 to 6.5 range, at which more nutrients become available to plants. However, it does not seem that the pH was a limitation in the present study, since all the media pH fell within the sufficiency range for the microgreen plants. Nevertheless, Ur Rahman et al. [33] reported that pH variation of the medium (from 5 to 9) significantly influenced the yield and biochemical constitutions in wheat (*Triticum aestivum* L.). The highest yield, total chlorophyll, and carotenoid contents were observed in seedlings grown in media with a neutral pH (6.5–7), while the lowest one was obtained in acidic (pH 5) and alkaline (pH 9) media that correspond with the results of this study.

Notably, there was a significant positive association between the yield, salinity, and TDS, suggesting sufficient growing medium fertility levels in particularly, T1.2 and T2.2, which were the only media with mushroom compost. Previous studies showed that high electric conductivity and salinity can reduce plant growth [34,35], which can be managed by adding perlite and wood-based substrates into the growing media to improve texture, structure, and porosity [35,36,37]. However, T1.2 and T2.2 had acceptable ranges of salinity thresholds between 640 and 1600 mg/L, as recommended for most vegetable crops [38]. Generally, NC recorded the highest salinity and the lowest yield, as previously reported by Shannon et al. [39], for kale and Swiss chard grown in media with excess salinity levels > 3.0 dS/m. Lin et al. [23] reported an increase in the plant growth and yield components of Swiss chard, pak choi, and kale in a medium consisted of 60% vermicast and 40% sawdust, with a considerably high electric conductivity of 1450 μS/cm and a pH of 7.3. Furthermore, Hernández et al. [27] attributed increased germination rate, fresh shoot weight, and yield in red baby leaf lettuce to mushroom compost, with a pH of 7 and an electric conductivity of > 4000 μS/cm. There was no significant correlation between EC and the measured growth components, but there was a strong relationship between pH and growth plant components in all the plants.

The microgreens’ biochemical composition was significantly altered by the different mixed growing media. There are very few documented reports on the effect of different mixed growing media on biochemical quality of microgreens. Previous studies have demonstrated that vermicast and mushroom compost are well known to be rich in macro- and micro-elements including N, which is essential for chlorophyll and carotenoid synthesis as well as photosynthesis [40,41]. In this study, total flavonoids and ascorbates ranged from 404.1 to 653.7 μg quercetin/g, and 18.1 to 30.9 μmol/g FW, respectively. Media T1.2 and T2.2 impacted the highest amount of microgreen flavonoids and ascorbate contents, respectively, that most likely can be associated with media nutrient availability and a balance in C/N ratio, due to the added mushroom compost as explained by Hernández et al. [27]. Moreover, it was demonstrated that mushroom compost may be chitin-rich, which can be a significant source of plant growth stimulants and elicitors for the biosynthesis of secondary metabolites [42,43]. Therefore, a significant amount of chitin might be present in T1.2 and T2.2, leading to the high microgreen plants content of total carotenoid, flavonoids, and ascorbate, compared to media without mushroom compost. Treatments T1.1 and T2.1 improved phenolics content in all the microgreens irrespective of plant species. This can be ascribed to the high-carbon input from the thermally treated sawdust and PittMoss. This carbon might have improved the carbon-based phenolic compounds and their precursors involved in plant defense mechanisms and responses to environmental stress [44]. Contrary to this, the total phenolics was lower in T2.2, which suggested that the probably high N content in T1.2 and T2.2, due to the addition of N-rich vermicast and mushroom compost, might have reduced phenolic content in the microgreens as previously reported [14,44,45]. The difference in growing media had a significant effect on POD and APEX enzymes activities in the microgreens. Several studies have reported a strong correlation between bioactive phytochemicals and antioxidant properties [10,46]. Besides the increased ascorbate and flavonoids contents, POD and APEX were highly increased in the microgreens grown in T1.2 and T2.2. Our results are consistent with findings obtained by Shiri et al. [19], who reported a significant increase in antioxidant capacity with an elevated ascorbic acid content in plants.

## 4. Materials and Methods

### 4.1. Plant Material and Growing Condition

The experiment was carried out in July 2020 and repeated in December 2020 in the Department of Plant, Food, and Environmental Sciences greenhouse (45°23′ N, 63°14′ W), Dalhousie University, Truro, NS, Canada. The microgreens were kale (*Brassica oleracea* L. var. *acephala*), Swiss chard (*Beta vulgaris* var. *cicla*), arugula (*Eruca vesicaria* ssp. *sativa*), and pak choi (*Brasica rapa* var. *chinensis*), purchased from Halifax Seed Co., Halifax, NS, Canada. The growing media were PittMoss, vermicast, sawdust, mushroom compost, perlite and Pro-mix BX™. PittMoss^®^ is a soilless potting mix made from recycled paper (Ambridge soil company, PA, USA). It is expected that the PittMoss will improve aeration and water retaining potential, resulting in the better delivery of nutrients to the root-zone environment. Vermicast, sawdust, and shiitake (Lentinula edodes) mushroom compost were obtained from Modgarden Company, Toronto, ON, Canada. Perlite and Pro-mix BX™ potting medium were purchased from Halifax Seed Company, NS, Canada. Kale, Swiss chard, arugula, and pak choi seeds were sown in flat plastic cell trays, measuring 19 cm length × 12 cm width × 2.5 cm deep, each containing a different mixed medium. The trays were kept in the greenhouse under a 16/8-hr day/night light regime (from high pressure sodium lamp) at a 24°/22 °C day/night temperature cycle with a 71% mean relative humidity. A 600 W HS2000 high-pressure sodium lamp with NAH600.579 ballast (P.L. Light Systems, Beamsville, ON, Canada) supplied the supplementary lighting. Air distribution in the greenhouse was distributed by a horizontal air-flow ventilation system. Watering was carried out every two days with 200 mL of tap water for each pot until the final harvest at 15 days after sowing. No additional fertilizer was applied.

### 4.2. Experimental Treatment and Design

The 2-factor experiment (i.e., plant species x growing media) was arranged in a completely randomized design with three replications. Seeds were sown in six different proportions of mixed media (Table 3). Pots were rearranged weekly on the growth shelf to offset microclimate variations in the greenhouse. The entire study was repeated twice. The data from the two studies were merged because the coefficient of variation was less than 5%. Seed germination, plant growth, yield, and various biochemical characteristics were measured.

### 4.3. Growing Media Physicochemical Properties

To evaluate chemical properties of the growing media, 50 g of each media was added to 50 mL of deionized water and was thoroughly mixed before the determination of chemical properties. pH, salinity, electrical conductivity, and total dissolved solids were measured using an ExStik^®^ II EC500 waterproof pH/conductivity meter (Extech ITM Instruments Inc., Newmarket, ON, Canada). The growing media physical properties and water retention characteristics were determined in triplicate as described by Armah [47], with slight modifications. Bulk density (D_b_) was determined from the weight (M) and volume (V_1_) of the soil core, using a graduated glass cylinder after continuous tapping, until there was no observable change in soil volume.
(1)Bulk density=MV1
(2)Porosity=MsV2

Water saturation, field capacity, and wilting point were determined after the soil was air-dried under ambient conditions (*ca*. 22 °C). A known mass of the fresh soil sample (M_s_) was placed in a 15.24 cm plastic pot with drainage holes and was weighed (M_sp_). The potted soil was placed in a saucer and was saturated with distilled water, and the saturated soil weight (M_sat_) was recorded after 48 h. Then, the saucer was removed so that the free water could drain out under atmospheric pressure for 72 h and was then weighed (M_drained_). The drained soil was spread evenly in a flat aluminum tray and air-dried under ambient conditions for 72 h and then weighed (M_dried_).
(3)$$Field\;capacity\;(F_{c})=\frac{M_{drained}-M_{sp}}{M_{s}}\times 100$$

### 4.4. Plant Growth and Yield Components

Data on seedling growth indices were collected 14 days after sowing the seeds. Plant samples (n = 15) were randomly and gently uprooted from the middle section of the growing trays for each treatment per replicate using a spatula. The seedlings were placed on tissue paper before carefully removing chunks of loosely attached media from the roots. The roots were then thoroughly washed under a gentle running deionized with minimum root loss (i.e., ca. < 2%). After drying with a blotting paper, the total lengths of roots and shoots and root volume were determined using a Perfection V800 Photo Color Scanner Digital ICE^®^ Technologies (Epson America Inc., Los Alamitos, CA, USA). The shoots of the remaining microgreens were cut with a pair of scissors at the growing media surface after 14 days of sowing, and the fresh weights were recorded as the estimated yield per treatment. At the final harvest, there was no seed residue on the shoots that we had to worry about.

### 4.5. Microgreen Quality and Phytochemical Analysis

#### 4.5.1. Chlorophylls a and b, Total Chlorophyll, and Total Carotenoid

Samples of the microgreens per treatment from the final harvest in Section 4.4 above were immediately frozen in liquid N to avoid changes in the biochemical compounds present in the plants. Pooled samples of the microgreens frozen in liquid N were ground to fine powder and stored in −20 °C until analyzed. Briefly, 0.2 g of each ground microgreen was separately dissolved in 10 mL of 80% acetone. After centrifuging at 12,000 rpm for 15 min, the supernatant was collected and transferred into 96 micro-well plates to measure the absorbance at 646.8 nm and 663.2 nm wavelength, using a UV-Vis spectrophotometer (Evolution™ Pro, Thermo Fisher scientific, Waltham, MA, USA) against acetone as blank, using the method described by Lightenthaler [48]. Chlorophyll and carotenoid concentrations were obtained by the following formula.
(4)Chla (µg/mL)=12.25×A663.2−2.79×A646.8
(5)Chlb (µg/mL)= 21.50×A646.8 – 5.1×A663.2
(6)Chlt (µg/mL)=chla+chlb
(7)Car (µg/mL)=(1000×A470−1.8×chla−85.02×chlb)/198

Finally, the calculated value was multiplied by the total volume (10 mL) and then divided by the total fresh weight (0.2 g), which was expressed as µg/g FW.

#### 4.5.2. Total Sugar

The total sugar content of the microgreens was measured using the method described by Mohammadkhani and Heidari [49], with some modifications. Firstly, 0.2 g of powder was dissolved in 10 mL of 90% ethanol and was incubated in a water bath for 60 min. The mixture was topped with up to 25 mL with 90% ethanol and centrifuged at 4000 rpm for 3 min. An amount of 1 mL of the supernatant was transferred into a glass test tube and 1 mL of 5% phenol was added and vortexed. Subsequently, 5 mL of sulfuric acid was added and incubated in the dark for 15 min. The mixture was cooled, and the absorbance was measured at 490 nm using a UV-Vis spectrophotometer against a blank made up of deionized water, phenol, and sulfuric acid. The total sugar was obtained by a standard sugar curve prepared by dissolving sucrose in distilled water at different concentrations, from 0 to 300 µg. Then, 1 mL of 5% phenol and 5 mL of sulfuric acid was added to the mixture and the absorbance was recorded at 490 nm. The sugar content was expressed as μg glucose/g FW.

#### 4.5.3. Total Protein

The total protein content was measured using the Bradford assay, as described by Hammond and Kruger [50]. In brief, 0.2 g of the ground microgreen tissue samples was transferred into a test tube, added with 5 mL ice-cold extraction buffer (i.e., 50 mM potassium phosphate buffer at pH 7.0) and 0.1 mM EDTA. The mixture was vortexed for 30 s before centrifugation at 15,000 rpm for 20 min. The supernatant was collected and kept on ice. Subsequently, the supernatant was mixed with 100 µL of enzyme extract and 1 mL of Bradford reagent, before recording the absorbance against a blank (Bradford reagent) at 595 nm after a 5 min incubation. The protein concentration was determined by the regression equation obtained from a Bovine serum albumin at different concentrations (200–900 µg mL^−1^) and was expressed as μg Bovine/g.

#### 4.5.4. Total Phenolics

The total phenolic (TPC) was measured using the Folin–Ciocalteu method, as described by Alothman et al. [51]. Briefly, 0.2 g of the ground microgreens was dissolved in ice-cold 80% methanol and incubated at an ambient temperature (approximately, 22 °C) for 48 h in the dark. The mixture was then centrifuged at 13,000 rpm for 5 min. A 100 μL sample of the supernatant, the standard at different concentrations (i.e., 0, 5, 10, 15, 20, 25 mg/L), and a methanol blank were added into distinct tubes before adding 200 μL Folin-Ciocalteu reagent and 800 μL of Na_2_CO_3_ and then incubating it for 2 h in the dark. Eventually, 200 μL of the mixture, the standard, and the blank were individually transferred into a microplate to measure the absorbance at 765 nm by UV-vis spectrophotometer. TPC concentration was determined by the standard curve obtained from Gallic acid equivalents and expressed as mM Gallic acid per g of fresh sample (mg GAE/g).

#### 4.5.5. Total Flavonoids

The total flavonoid was measured using the method described by Chang et al. [25]. Ground samples of each microgreen (0.2 g) and 2.5 mL of 95% methanol was mixed and vortexed before centrifugation at 13,000 rpm for 10 min. The supernatant (500 μL) standard (1 mg quercetin dissolved in 95% methanol at 5, 10, 15, 25, 50, 100, 150, 200 µg/mL concentrations), and 95% methanol were transferred into separate tubes. Then, 1.5 mL 95% methanol, 0.1 mL 10% AlCl_3_, 0.1 mL 1 M potassium acetate, and 2.8 mL distilled water were added to each tube. Afterward, the mixture was incubated at an ambient temperature for 30 min, and the absorbance was recorded at 415 nm against a blank using a UV-Vis spectrophotometer. The flavonoids content was measured by the standard curve obtained from the quercetin standard curve. The total flavonoids content was expressed as μg quercetin/g of plant fresh weight.
(8)$$\mathrm{Total}\;\mathrm{flavonoid}=\frac{\left(\left[flavonoids\right]\left(µg/mL\right)\times total\;volume\;of\;methanolic\;extract\;\left(mL\right)\right)\;\;}{mass\;of\;extract\;\left(g\right)\;}$$

#### 4.5.6. Total Ascorbate

The total ascorbate was measured using the method described by Ma et al. [52]. In brief, 0.2 g of the ground microgreens was mixed with 1.5 mL ice-cold 5% trichloroacetic acid (TCA) and centrifuged for 15 min at 4 °C. Then, 100 μL of the supernatant was collected and added to 400 μL phosphate buffer (150 mM KH_2_PO_4_), 5 mM EDTA, and 100 μL10 mM dithiothreitol and vortexed. Following the incubation of the mixture, 0.5% N-ethylmaleimide was added to the mixture and vortexed. To obtain color, 400 μL 10% TCA, 400 μL 44% orthophosphoric acid, 400 μL4% dipyridyl and 200 μL 30 g/L FeCl_3_ was added to the mixture and incubated at 40 °C for 1 h before recording the absorbance at 525 nm using a UV-Vis spectrophotometer against a blank. The standard was prepared from L-ascorbic acid in 5% TCA (0–5 mM). Total ascorbate content was expressed as μmol/g FW.

#### 4.5.7. Antioxidant Enzyme Activity

The peroxidase (POD) and ascorbate peroxidase enzyme activities (APEX) were measured using the method described by Patterson et al. [53]. Briefly, 0.2 g of the ground microgreens was mixed with 5 mL ice-cold extraction buffer and centrifuged at 15,000 rpm for 20 min. The extraction buffer contained mM potassium_–_phosphate buffer (pH 7.0), 1% polyvinylpyrrolidone, and 0.1 mM EDTA. The supernatant (i.e., enzyme extract) was collected for POD and APEX assays. For POD, the reaction mixture was prepared from the combination of 100 mM potassium-phosphate buffer (pH 7.0), 0.1 mM pyrogallol, and 5 mM H_2_O_2_. Then, 10 μL of the supernatant was added to the mixture and incubated for 5 min at room temperature. To stop any enzyme reaction in the mixture, 0.1 mL of NH_2_SO_4_ was added. Finally, the absorbance was recorded at 420 nm using a UV-Vis spectrophotometer against a blank (Milli-Q water). The enzyme activity was calculated by the following formula and expressed as unit/mg FW.
POD = A_420_ × 3/(12 × 0.1))/0.2(9)

To assay APEX, 100 μL of the supernatant was added to the reaction mixture, i.e., 1372 μL of 50 mM potassium_–_phosphate buffer (pH 7.0), 75 μL of 10 mM ascorbate, and 3 μL of 100 mM H_2_O_2_. The mixture was incubated for 1 min before reading the absorbance at 290 nm using a UV-Vis spectrophotometer against a blank. The enzyme activity in unit/mg FW was obtained by:APEX = (A 290 × 1/(2.8 × 0.1))/0.2(10)

### 4.6. Statistical Analysis

All the data were subjected to a two-way analysis of variance (ANOVA) using Minitab version 18.3. Fisher method was used to separate treatment means when the ANOVA showed a significant difference at *p* < 0.05. Furthermore, a multivariate analysis using a two-dimensional principal component analysis (PCA) was carried out using GenStat software.

## 5. Conclusions

Global warming and climate change have had adverse impacts on plant production and food security. During the last decade, synthetic chemical fertilizers and pesticides have been extensively used in conventional agriculture to meet global food and nutrition demand. However, their application negatively affects the environment and human health. Therefore, the development of an innovative and climate-smart approach to food production is of high importance. In the present study, the effect of different mixed natural growing media on the growth and biochemical properties of different microgreen plant species was investigated. Overall, our results showed that variations in the growing media characteristics had a significant effect on the studied traits of the microgreens. Overall, growing media containing mushroom compost, i.e., T2.2, was found to be the most favorable. The efficacy of T2.2 on the assessed growth, yield, and quality traits was further confirmed through the PCA analysis. The ingredients used to make the mixed growing media in this study are reasonably inexpensive and locally available. Therefore, they can be used as an alternative to conventional media such as Pro-mix BX™ potting medium for growing microgreens to improve productivity and nutrient and non-nutrient bioactive compounds.

## Figures and Tables

**Figure 1 plants-11-03546-f001:**
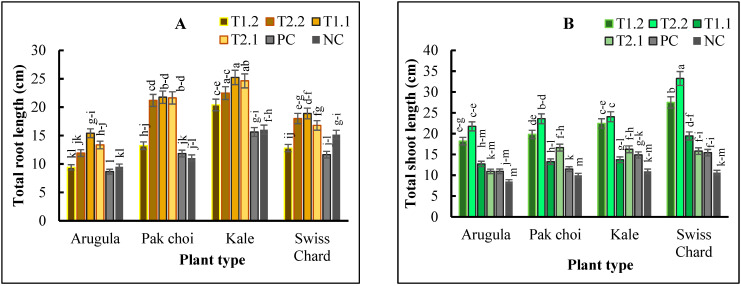
Total root length (**A**); total shoot length (**B**); root volume (**C**); yield (**D**) of arugula (*Eruca vesicaria* ssp. sativa), pak choi (*Brasica rapa* var. chinensis), kale (*Brassica oleracea* L. var. acephala) and Swiss chard (*Beta vulgaris* var. cicla) microgreens as affected by different growing media comprised of T1.1: 30% vermicast + 30% sawdust + 10% perlite + 30% PM; T1.2: 30% vermicast + 30% sawdust + 10% perlite + 30% MC; T2.1: 30% vermicast + 20% sawdust + 20% perlite + 30% PM; T2.2: 30% vermicast + 20% sawdust + 20% perlite + 30% MC; NC: 60% sawdust + 40% PittMoss; and PC: Pro-mix BX™ potting medium alone. Vertical bars represent standard errors of the means (N = 3). Bars with a common lower-case letter signifies treatment means that were not significantly different at *p* < 0.05.

**Figure 2 plants-11-03546-f002:**
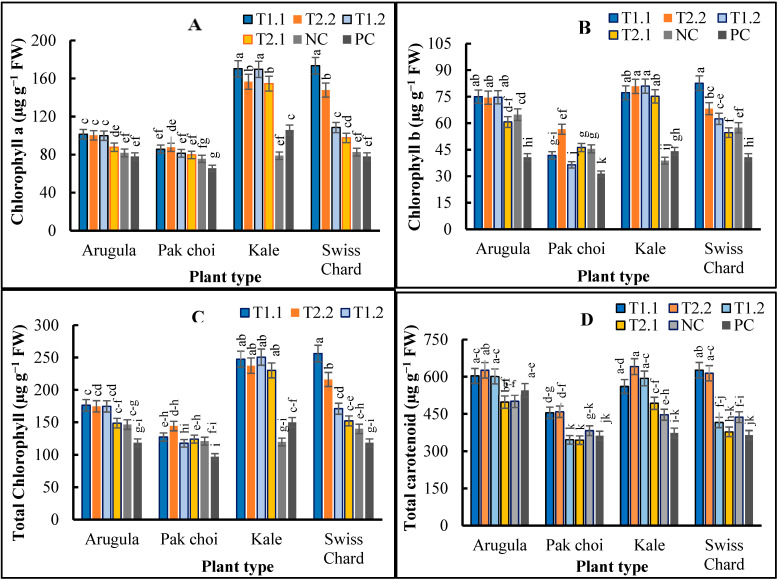
Chlorophyll a (**A**) and b (**B**), total chlorophyll (**C**) and carotenoid (**D**) contents of arugula (*Eruca vesicaria* ssp. sativa), pak choi (*Brasica rapa* var. chinensis), kale (*Brassica oleracea* L. var. acephala) and Swiss chard (*Beta vulgaris* var. cicla) microgreens as affected by different growing media comprised of T1.1: 30% vermicast + 30% sawdust + 10% perlite + 30% PM; T1.2: 30% vermicast + 30% sawdust + 10% perlite + 30% MC; T2.1: 30% vermicast + 20% sawdust + 20% perlite + 30% PM; T2.2: 30% vermicast + 20% sawdust + 20% perlite + 30% MC; NC: 60% sawdust + 40% PittMoss; and PC: Pro-mix BX™ potting medium alone. Vertical bars represent standard errors of the means (N = 3); significant at *p* < 0.01. Bars with a common lower-case letter signifies treatment means that were not significantly different at *p* < 0.05.

**Figure 3 plants-11-03546-f003:**
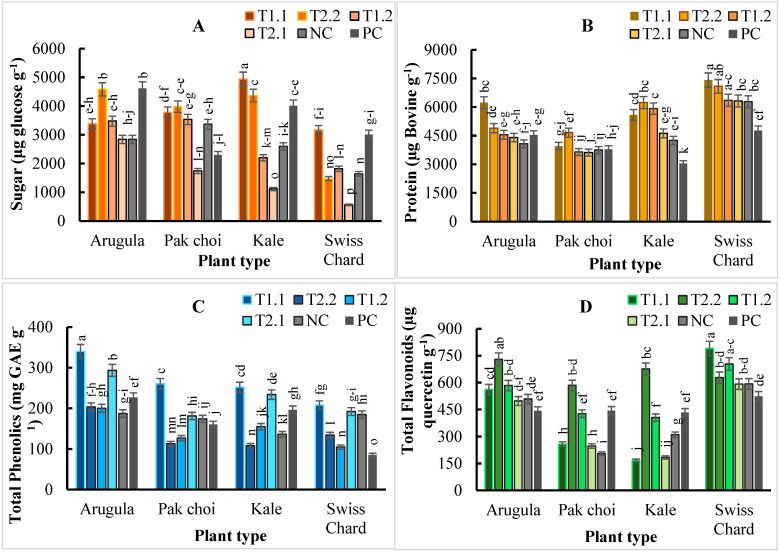
Sugar (**A**); protein (**B**); total phenolics (**C**); total flavonoids (**D**) contents of arugula (*Eruca vesicaria* ssp. sativa), pak choi (*Brasica rapa* var. chinensis), kale (*Brassica oleracea* L. var. acephala) and Swiss chard (*Beta vulgaris* var. cicla) microgreens as affected by different growing media comprised of T1.1: 30% vermicast + 30% sawdust + 10% perlite + 30% PM; T1.2: 30% vermicast + 30% sawdust + 10% perlite + 30% MC; T2.1: 30% vermicast + 20% sawdust + 20% perlite + 30% PM; T2.2: 30% vermicast + 20% sawdust + 20% perlite + 30% MC; NC: 60% sawdust + 40% PittMoss; and PC: Pro-mix BX™ potting medium alone. Vertical bars represent standard errors of the means (N = 3); significant at *p* < 0.01. Bars with a common lower-case letter signifies treatment means that were not significantly different at *p* < 0.05.

**Figure 4 plants-11-03546-f004:**
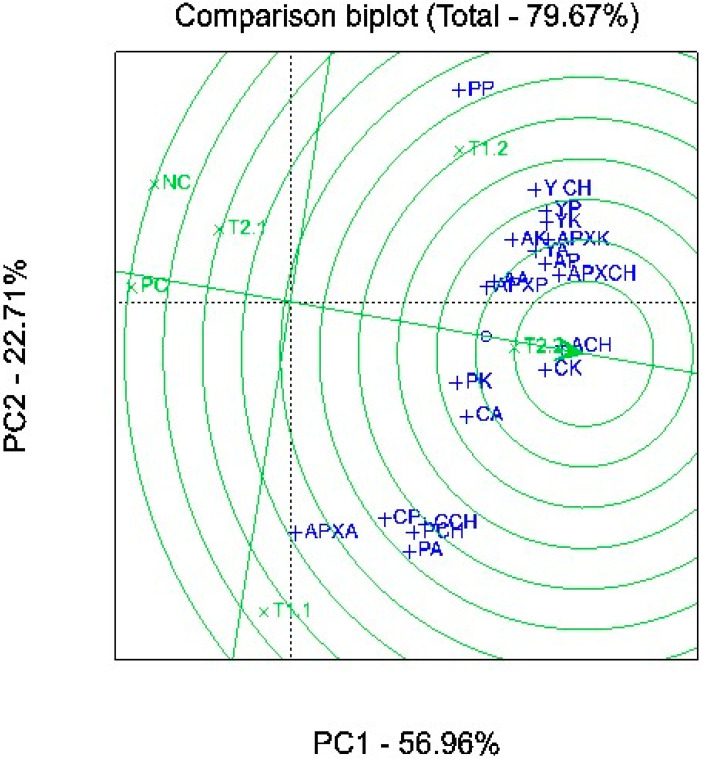
Ranking total × total biplot for comparison of treatment × plant species interaction effects on biochemical variations in all microgreens. Arugula (*Eruca vesicaria* ssp. sativa) yield (YA), arugula ascorbate (AA), arugula carotenoids (CA), arugula POD Activity (PA), arugula APEX Activity (APXA); Swiss chard (*Beta vulgaris* var. cicla) yield (YCH), Swiss chard ascorbate (ACH), Swiss chard carotenoids (CCH), Swiss chard POD Activity (PCH), Swiss chard APEX Activity (APXCH); kale (*Brassica oleracea* L. var. acephala) yield (YK), kale ascorbate (AK), kale carotenoids (CK), kale POD Activity (PK), kale APEX Activity (APXK); pak choi (*Brasica rapa* var. chinensis) yield (YP), pak choi ascorbate (AP), pak choi carotenoids (CP), pak choi POD Activity (PP), pak choi APEX Activity (APXP). T1.1: 30% vermicast + 30% sawdust + 10% perlite + 30% PM; T1.2: 30% vermicast + 30% sawdust + 10% perlite + 30% MC; T2.1: 30% vermicast + 20% sawdust + 20% perlite + 30% PM; T2.2: 30% vermicast + 20% sawdust + 20% perlite + 30% MC; NC: 60% sawdust + 40% PittMoss; and PC: Pro-mix BX™ potting medium alone.

**Table 1 plants-11-03546-t001:** Physiochemical properties of growing media affected by different proportions of mixed amended.

Treatment	Bulk Density (g/cm^3^)	Porosity (%)	Field Capacity (%)	pH	Salinity (mg/L)	Electric Conductivity (μS/cm)	Total Dissolved Solids (mg/L)
T1.1	0.07 b	31.3 c	29.2 c	5.7 b	1299.7 c	2260.0 c	1719.6 c
T1.2	0.12 a	35.0 b	34.2 a	6.4 a	1689.7 a	2570.0 b	2139.7 b
T2.1	0.07 b	26.6 d	25.5 d	5.8 ab	1319.7 c	1233.0 e	1709.6 c
T2.2	0.10 ab	35.7 b	33.2 ab	6.3 ab	1494.9 b	2205.5 c	2028.4 b
PC	0.10 ab	37.8 a	30.2 bc	6.1 ab	802.9 d	1486.0 d	1233.5 d
NC	0.09 ab	27.6 d	24.5 d	5.9 ab	1861.2 a	3412.5 a	2479.7 a
*p*-value	0.015	0.000	0.000	0.029	0.000	0.001	0.001

T1.1: 30% vermicast + 30% sawdust + 10% perlite + 30% PittMoss (PM); T1.2: 30% vermicast + 30% sawdust + 10% perlite + 30% mushroom compost (MC); T2.1: 30% vermicast + 20% sawdust + 20% perlite + 30% PM; T2.2: 30% vermicast + 20% sawdust + 20% perlite + 30% MC; negative control (NC): 60% sawdust + 40% PittMoss; and positive control (PC): Pro-mix BX™ potting medium alone; significant at *p* < 0.05. Treatment means followed by a common letter are not significantly different.

**Table 2 plants-11-03546-t002:** The effects of mixed growing media on total ascorbate, peroxidase activity and ascorbate peroxidase activity.

Treatment	Total Ascorbate (μmol g^−1^ FW)				Peroxidase Activity (Unit mg^−1^ FW)				Ascorbate Peroxidase Activity (Unit mg^−1^ FW)			
	Arugula	Pak Choi	Swiss Chard	Kale	Arugula	Pak Choi	Swiss Chard	Kale	Arugula	Pak Choi	Swiss Chard	Kale
T1.1	24.0 de	20.4 fgh	25.4 cd	23.6 de	0.94 bc	0.50 ij	0.96 bc	0.56 gh	0.23 a	0.08 fg	0.07 fg	0.06 g
T2.2	24.2 cde	28.0 bc	32.0 a	29.7 ab	0.67 ef	1.05 b	1.02 b	0.54 hij	0.15 cd	0.07 g	0.19 ab	0.19 ab
T1.2	32.2 a	29.5 ab	30.0 ab	32.0 a	0.60 fgh	1.25 a	0.50 ij	0.64 f	0.11 e	0.11 e	0.16 bcd	0.17 bcd
T2.1	21.5 efg	22.4 ef	21.9 ef	28.9 b	0.50 ij	0.94 bc	0.40 k	0.48 j	0.18 abc	0.04 j	0.11 e	0.05 hi
NC	17.9 hij	16.0 j	18.6 ghij	19.8 fgh	0.29 k	0.75 de	0.51 ij	0.17 m	0.06 g	0.06 gh	0.07 j	0.08 f
PC	20.6 fgh	18.1 hij	17.4 ij	21.4 efg	0.50 ij	0.87 cd	0.62 fg	0.50 ij	0.15 d	0.06 gh	0.05 de	0.07 fg
*p* value G	0.001	0.000	0.000	0.000	0.001	0.001	0.000	0.000	0.000	0.000	0.000	0.000
P	0.000	0.000	0.000	0.000	0.000	0.000	0.001	0.001	0.000	0.000	0.000	0.000
G × P	0.000	0.000	0.000	0.000	0.000	0.000	0.000	0.000	0.000	0.000	0.000	0.000

T1.1: 30% vermicast + 30% sawdust + 10% perlite + 30% PM; T1.2: 30% vermicast + 30% sawdust + 10% perlite + 30% MC; T2.1: 30% vermicast + 20% sawdust + 20% perlite + 30% PM; T2.2: 30% vermicast + 20% sawdust + 20% perlite + 30% MC; negative control (NC): 60% sawdust + 40% PittMoss; and positive control (PC): Pro-mix BX™ potting medium alone.; significant at *p* < 0.01. Treatment means followed by a common letter are not significantly different. G, growing media; P, plant species; G × P, interaction of growing media and plant species (N = 4).

**Table 3 plants-11-03546-t003:** Proportions of mixed growing media.

Treatment	Formulation
T1.1	30% vermicast + 30% sawdust + 10% perlite + 30% PittMoss (PM)
T1.2	30% vermicast + 30% sawdust + 10% perlite + 30% mushroom compost (MC)
T2.1	30% vermicast + 20% sawdust + 20% perlite + 30% PittMoss (PM)
T2.2	30% vermicast + 20% sawdust + 20% perlite + 30% mushroom compost (MC)
NC	60% sawdust + 40% PittMoss
PC	Pro-mix BX™ potting medium alone

NC and PC are negative and positive control, respectively.

## Data Availability

Not applicable.

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
