# Peer review of "Growth and Biochemical Composition of Microgreens Grown in Different Formulated Soilless Media"

_plants, 2022, doi:10.3390/plants11243546_

Round 1
Reviewer 1 Report
The paper is good, however I would suggest some improvement regarding certain points.
1. The paper is too long, too much data are presented. It is more of a scientific thesis than an article. I belive that a strong shrinking of that would make a benefit.
2. The materials and methods should focus more on the practical parts. Eg. the "watering was done every two days (or when required)" would need to be described a little bit more precisely, since water supply in such cases may be fundamental regarding growing conditions.
3. Less information is provided in the field of pH. A most crucial characteristc of any growth medium is the alkalo-acidity state of that, and even the changes during the growing period. If there are no data about, it will have to be introduced and discussed . maybe by literature citations.
4. The figures are too small, too many of them and sometimes the abundance drives the reader to confusion.
So, altogether I think this paper is typical "the less would have yielded more".
Author Response
Dear Reviewer,
On behalf of my co-authors, I would like to express my gratitude for your useful comments and suggestions to improve our manuscript.
We have incorporated all your suggestions and the detail response is attached.
Thanks,

Reviewer 2 Report
The paper “Growth and Biochemical Composition of Microgreens Grown in different Formulated Soilless Media” is a well written paper. The objective of the present study was to evaluate different mixed growing media on growth, chemical composition, and antioxidant activities of four microgreen species (kale, Swiss chard, arugula and pak choi) that can be grown and harvested as immature greens.
The introduction is informative and covers all the needed aspects for the growing media and microgreens. The 55 references covered spherically all the bibliography.
Only the following issues need more details:
Line 392: what about the relative humidity and air distribution in the greenhouse?
Line 393: What is the efficacy of the used lamp?
Also I think that the writers in the conclusion session has to mention sth about the cost of this growing media which contained natural ingredients in comparison with the current used ones and also about their availability in the market.
Author Response
Dear Reviewer,
On behalf of my co-authors, I would like to express my gratitude for your comments and suggestions to improve the quality of our manuscript.
We have attached the detailed response to your comments/suggestions here.
Thanks,
Lord
